# Photocatalytic Applications of ReS_2_-Based Heterostructures

**DOI:** 10.3390/molecules28062627

**Published:** 2023-03-14

**Authors:** Nan Wang, Yashu Li, Lin Wang, Xuelian Yu

**Affiliations:** Engineering Research Center of Ministry of Education for Geological Carbon Storage and Low Carbon Utilization of Resources, Beijing Key Laboratory of Materials Utilization of Nonmetallic Minerals and Solid Wastes, National Laboratory of Mineral Materials, School of Materials Science and Technology, China University of Geosciences (Beijing), Beijing 100083, China

**Keywords:** ReS_2_, transition metal dichalcogenides, heterostructure, photocatalysis, photocatalytic hydrogen evolution, photocatalytic CO_2_ reduction, photocatalytic pollutant degradation

## Abstract

ReS_2_-based heterostructures, which involve the coupling of a narrow band-gap semiconductor ReS_2_ with other wide band-gap semiconductors, have shown promising performance in energy conversion and environmental pollution protection in recent years. This review focuses on the preparation methods, encompassing hydrothermal, chemical vapor deposition, and exfoliation techniques, as well as achievements in correlated applications of ReS_2_-based heterostructures, including type-I, type-II heterostructures, and Z-scheme heterostructures for hydrogen evolution, reduction of CO_2_, and degradation of pollutants. We believe that this review provides an overview of the most recent advances to guide further research and development of ReS_2_-based heterostructures for photocatalysis.

## 1. Introduction

Currently, the need for clean and renewable energy sources in order to mitigate the increasing environmental pollution and global energy crisis has become more urgent. Photocatalytic technology has emerged as a promising avenue for utilizing solar energy due to its low cost, abundance, and eco-friendly nature [1]. In 1972, Fujishima and Honda demonstrated for the first time the ability to produce hydrogen by splitting water on a TiO_2_ electrode [2]. Since then, TiO_2_ has been extensively studied due to its strong redox abilities, eco-friendliness, and chemical stability, which have made it a promising material for photocatalysis over the past few decades [3,4,5,6]. Despite its advantages, TiO_2_ suffers from a narrow spectral response range, a high carrier recombination rate, and a lack of catalytic active sites, thus limiting its photocatalytic hydrogen evolution (PHE) activity [7,8,9]. The emergence of two-dimensional (2D) photocatalysts, such as transition metal dichalcogenides (TMDs), has reinvigorated the progress in photocatalytic and photoelectrocatalytic research [10,11,12]. Due to their easy exfoliation into monolayers, TMDs provide numerous benefits such as a high surface area and decreased migration distance for photogenerated electron–hole pairs. This not only offers an increased number of reaction sites but also reduces the likelihood of electron–hole recombination, potentially improving photocatalytic performance [13]. Nevertheless, the electronic structure of TMDs is highly dependent on the number of layers [14]. Achieving precise control of the number of layers is critical to obtaining optimal catalytic performance, as exemplified by the fact that the photocatalytic hydrogen production of MoS_2_ is substantially improved when it converts from a multi-layer to a single-layer configuration [15]. As such, controlling the number of layers remains a daunting task that limits the utility of TMDs in photocatalytic applications [16,17].

As a newly emerged member of the 2D TMDs family, ReS_2_ has been widely studied in the fields of electronics [18,19], optoelectronics [20,21,22,23,24], and catalysis [25,26,27,28] due to its advantageous features such as a large surface area, tunable active sites, layer-independent electric/optical properties, and structural/vibrational anisotropy. ReS_2_ crystallizes in a twisted T phase with clusters of Re_4_ units, forming one-dimensional chains in each single layer [29]. Distinct from the conventional 2H phase TMDs materials (represented by MoS_2_ in Figure 1a) with high hexagonal symmetry, ReS_2_ belongs (Figure 1b) to the triclinic crystal system with only one symmetrical center, exhibiting a distorted 1T’ phase structure. The Re atoms show an octahedral coordination. The extra valence electron on the 5d orbital of Re results in a Peierls distortion, and the adjacent four Re atoms are linked by metallic bonds to form a Re_4_ cluster, which further distorts the original hexagon of ReS_2_ into a parallelogram [30]. It has been demonstrated that bulk ReS_2_ consists of electronically and dynamically decoupled layers bound by weak van der Waals forces, which gives rise to a pseudo single-layer architecture with nearly identical band structures and Raman active modes compared to that of its monolayer [29,31], consequently overcoming the difficulty of preparing large-area and single-crystal monolayers. Distinct from the indirect-to-direct band-gap transition of other group-VI TMDs, both single-layer ReS_2_ and multilayer ReS_2_ are shown to have a direct band gap of approximately 1.4 eV (Figure 1c), with suitable band-edge positions for photocatalytic water splitting [29,32]. Additionally, the weak interlayer coupling allows for easy preparation of nanosheets with ample surface area for co-catalysis with other materials. Lastly, ReS_2_ is significantly more adept at light absorption and carrier mobility than group-VI TMDs, thus making it an advantageous and effective photocatalyst under visible light illumination [32].

In 2018, Fu’s group undertook an experimental study on the optoelectronic properties of ReS_2_ and its performance in photocatalyzed hydrogen evolution [33]. The conduction and valence band edges of ReS_2_ are more closely spaced to the water redox potentials than CdS, thus rendering ReS_2_ more efficient for PHE. Moreover, the monolayer-like structure of the catalyst facilitates the formation of trions (a bound state of two electrons and one hole). These trions can then migrate to the surface and participate in two-electron reactions at the abundant active sites. Furthermore, the relatively intact basal planes of ReS_2_ can promote electron transport to the edge-reactive sites. Such a two-electron catalytic reaction endows ReS_2_ with a PHE rate of 13.023 mmol g^−1^ h^−1^ under visible light irradiation [34].

ReS_2_ is a versatile material but relatively less-explored among TMDs. It has garnered significant attention from the scientific community due to its unique properties compared to other TMDs. Rahman et al. critically analyzed the synthesis, structure growth mechanisms, properties, and contemporary applications of ReS_2_ in different areas [32]. Zhang et al. has highlighted the weak coupling characteristics of ReS_2_ [35], and the intrinsic anisotropic properties are summarized by Cao et al. [36]. Other recent reviews have summarized advances in ReS_2_ research with respect to electronics, optics, magnetism, and alkali-metal ion batteries [37,38,39,40]. Academic research regarding the photocatalytic properties of ReS_2_ materials is currently limited.

It is well established that heterostructures have numerous advantages in enhancing photocatalyst functions [1,41,42]. Rational alignment of ReS_2_ with other semiconductors into a tandem structure can increase the range of light absorption across the solar spectrum and optimize its photocatalytic performance by inhibiting charge carrier recombination as well as promoting charge separation [34,43,44,45,46]. In this review, we provide mainly an overview of the recent progress on ReS_2_ heterostructure, focusing on the synthesis methods and their photocatalytic applications. Firstly, the preparation methods are discussed, including hydrothermal reaction, chemical vapor deposition (CVD), and mechanical or chemical exfoliation. Furthermore, we focus mainly on photocatalytic applications based on ReS_2_-based heterostructure, such as photocatalytic hydrogen evolution, photocatalytic reduction of CO_2_, and pollutant degradation. Finally, the review is summarized, and the prospects are discussed from current views.

## 2. Synthetic Methods for ReS_2_ and Its Composite Material

Various synthesis techniques such as mechanical/chemical exfoliation, hydrothermal, and CVD methods have been developed to prepare ReS_2_-based photocatalysts with different morphologies including nanospheres, nanosheets, etc. [45,47], all of which can be used as core components in the construction of photocatalytic heterostructures.

### 2.1. Exfoliation from the Bulk ReS_2_

The weak interlayer coupling of ReS_2_ permits the separation of bulk ReS_2_ into nanosheets through exfoliation, which is a simple, high-yield method for ReS_2_ preparation [29,48,49]. Ultrathin nanostructures can be obtained by employing a hybrid of mechanical and liquid exfoliation. Bulk ReS_2_ produced using either commercial or hydrothermal synthesis can be dispersed in organic solvents (such as ethanol or N,N-dimethylformamide) through ultrasound-assisted liquid-phase exfoliation at 20–25 °C using an ice bath, followed by centrifugation and filtration to yield the nanosheets [34,47,50,51,52,53]. The bulk ReS_2_ can be exfoliated into a few layers (Figure 2a), and the original structure of ReS_2_ is maintained. Figure 2b,c show the intact basal plane and edge of ReS_2_ nanosheets, respectively [50]. Figure 2d,e display the ReS_2_ nanosheets with a thickness of about 4.20 nm [54]. The under-coordinated sulfur sites at the edges of ReS_2_ nanosheets serve as highly active centers for H_2_ generation, while the intact basal plane facilitates efficient electron transport to these edge S active sites for H_2_ evolution [34]. Chemical methods, such as the addition of diazonium salts, can also be employed to modify the surface structure of TMDs during the stripping process [55]. For example, the incorporation of hydroxyl functional groups such as carboxyl can enhance the hydrophilicity of ReS_2_ and improve its dispersibility in the ReS_2_−C_6_H_5_COOH/TiO_2_ physical mixture, thus preserving the photocatalytic active sites [56].

### 2.2. Hydrothermal Synthesis Reaction

The hydrothermal method is one of the simplest methods for the preparation of TMDs [16]. Specific precursors are used to carry out chemical reactions under high temperature and high pressure in a specially designed polytetrafluoroethylene-lined autoclave. For the synthesis of pristine ReS_2_, ammonium perrhenate (NH_4_ReO_4_), hydroxylamine hydrochloride (NH_2_OH·HCl), and thiourea (CS(NH_2_)_2_) are dissolved in deionized water. After stirring, they are transferred to a polytetrafluoroethylene autoclave for hydrothermal reaction by heating to 200–240 °C for 12–48 h [58,59,60]. After cooling, the ReS_2_ black powder is repeatedly washed with deionized water and ethanol, and the final product is collected after drying. The ReS_2_ prepared using this method typically displays the morphology of nanospheres, and the size of the ReS_2_ nanospheres can be adjusted by regulating the concentration of rhenium sources in the reaction [45].

The enhanced photocatalytic activity of ReS_2_ can be achieved by combining it with another semiconductor material to construct heterojunctions [61]. There are two ways to prepare ReS_2_-based heterojunctions via the hydrothermal method. The first is to incorporate the synthesized host semiconductor material into the precursor mixture for the preparation of ReS_2_, or vice versa, using a two-step hydrothermal method [46,62]. Wang et al. dispersed the generated TiO_2_ nanofibers into the ReS_2_ preparation solution for the hydrothermal reaction to anchor the ReS_2_ nanosheets on the porous TiO_2_ nanofibers to form TiO_2_@ReS_2_ composites. The few-layer ReS_2_ nanosheets with twisted 1T phase that were prepared via the hydrothermal method show large catalytic active sites, high electron mobility, and matching band structure [58]. The other is to fabricate a heterostructure with all the raw material in the one-pot hydrothermal method [63]. Chen et al. prepared a 1T’-ReS_2_/g-C_3_N_4_ heterostructure via the one-pot solvothermal reaction (Figure 2f). The obtained 1T’-ReS_2_ nanosheets can be converted to 2H-ReS_2_ after subsequent heat treatment (Figure 2g). The thin 1T’-ReS_2_ nanosheets with multi-layers grown on the g-C_3_N_4_ nanotubes (green circle in Figure 2h) show characteristic 2D-layered structures of TMDs with the single-layer thickness ~0.27 nm and the interlayer spacing ~0.65 nm (blue circle in Figure 2h). Alternatively, ReS_2_ prepared via the hydrothermal method can also be subsequently combined with other semiconductor materials through grinding or liquid-phase dispersion, thereby enabling the construction of heterojunctions [51,64].

### 2.3. Chemical Vapor Deposition Technique

ReS_2_ nanosheets prepared through hydrothermal and exfoliation methods contain numerous impurities that can act as electron-and-hole recombination centers, thus reducing the photocatalytic activity of ReS_2_ [65]. Compared to other methods, CVD-prepared TMDs have many advantages, including uniform appearance, controllable thickness, low defect rate, clean interface, and high synthesis efficiency [66,67]. On the one hand, the weak van der Waals interaction between TMDs makes it possible to stack them to form a variety of vertical heterostructures. On the other hand, chemical bonds can be formed at the interface to realize the interlocking of different 2D materials to form horizontal heterostructures with special interfacial properties. Vertical heterostructures can be prepared through methods such as mechanical exfoliation and transfer [68,69], direct CVD growth [70], electrochemical exfoliation [71], and liquid-phase exfoliation [47]. Horizontal heterostructures can generally be prepared only by the CVD approach [72,73,74].

The CVD preparation of ReS_2_ heterostructures can also be divided into a “one-step method” and a “two-step method”. The one-step method puts the precursor containing the elements of the synthesized heterostructure into the reaction system at one time. This method is based on the difference in the substrate adsorption energy or the control of growth conditions, such as growth temperature or carriers, to realize the growth of two materials one after another. For example, by utilizing the difference in adsorption energy between Re and W atoms on a Au substrate, a one-step synthesis of a vertical ReS_2_/WS_2_ heterojunction on a Au/W-Re alloy foil can be achieved [75]. The two-step method means that the growth of heterostructures is carried out in two separated steps. These methods can realize the high-quality and large-area preparation of single-crystal ReS_2_, which has been widely used in micro-nano electronic devices and photoelectric detection fields [22,23,76]. A substantial amount of work has reported the CVD preparation and growth mechanism of ReS_2_ heterostructure; readers can refer to the related papers [72,73,74,75,77,78] and reviews [79].

The majority of TMDs produced via CVD techniques exhibit a parallel growth relative to the underlying substrate, resulting in a flat film morphology. Under similar CVD growth conditions, ReS_2_ nanosheets display a propensity to bend and adopt a vertically oriented configuration on the underlying substrate [80]. Specifically, the vertical growth of two-dimensional ReS_2_ is highly spontaneous and independent of the substrate [81]. The high packing density (∼10^8^ cm^−2^) of the ReS_2_ nanosheets promotes a substantial number of active Re atoms to be exposed at the sheet edges, providing abundant active sites for catalytic processes [33,70,80,82,83,84]. The common precursors for the CVD synthesis of ReS_2_ are ammonium rhenate (NH_4_ReO_4_), Re powder [82], and rhenium-containing oxides (ReO_3_ or Re_2_O_7_), with sulfur powder used as the predominant sulfur source and heated in a different zone [33,85,86]. The ReS_2_ prepared via the CVD method exists mostly in petal-like ReS_2_ nanowalls arranged vertically without agglomeration [28,33,87]. Zhang et al. anchored the ReS_2_ nanowalls vertically on the surface of the Si/SiO_2_ substrate (Figure 2i), which can distribute ultra-uniformly with a dense structure over a wide area (Figure 2j) [33]. The average length of the ReS_2_ nanowalls can be up to 200 nm with domain sizes of about 10 layers. The interplanar distance of the ReS_2_ nanowalls is about 0.62 nm (Figure 2k), corresponding to the (001) crystal plane. The beneficial exposure of the (001) facet ensured that more active sites were accessible for hydrogen adsorption, consequently enhancing the catalytic activity. The growth of the ReS_2_ single-crystal nanosheets does not need a polished substrate or a small lattice mismatch [87]. Cheng et al. grew ReS_2_ tightly on the TiO_2_-coated b-Si substrate with the shape of nanopyramids via the CVD method. The obtained ReS_2_ nanosheet was of high quality and exhibited a good crystal structure without defects [28]. Before the characterization of photocatalytic properties, ReS_2_ generally needs to be scraped off from the substrate, and then via the solution reaction (such as hydrothermal or self-assembly) to prepare heterostructures.

## 3. Heterostructured ReS_2_ Composites

The heterojunctions formed between the host semiconductors generate an internal electric field that facilitates the separation of electron–hole pairs and accelerates the migration of carriers, which is constructive for improving the photocatalytic efficiency [88,89,90]. ReS_2_-based heterostructures can fulfill the aforementioned requirements and demonstrate outstanding photocatalytic performance. According to the bandgaps and the electronic affinity of semiconductors, ReS_2_-based heterostructures can be divided into three different types: type-I (straddling gap), type-II (staggered gap), and Z-scheme system, the band alignment of which is shown in Figure 3. The band gap, the electron affinity, and the work function of the combined semiconductors can determine the dynamics of the electron and hole in the semiconductor heterojunctions.

### 3.1. Type I Heterostructure

In a type-I heterostructure (shown in Figure 3a), the photo-excited electrons and holes in the wide-gap material transfer to the narrow-gap material (ReS_2_), as demonstrated by the arrows [46,50,51,53,54,91,92,93]. The quantum confinement of the accumulated electrons and holes in the same semiconductor facilitates high carrier recombination rate and radiative recombination, which is incredibly detrimental to photocatalytic activity but desirable in light-emitting applications [91,94]. To inhibit recombination of photo-generated charge, a cocatalyst such as Pt that is deposited in situ onto the surface [54] or abundant sulfur vacancies [93] can act as an electron trapper to facilitate electron accumulation and transfer. Additionally, the abundant edge reaction sites in ultrathin ReS_2_ nanosheets in the composites can also accumulate electrons, thus accelerating reaction kinetics of the hydrogen evolution [53].

### 3.2. Type II Heterostructure

In a type-II heterojunction semiconductor, both valence band (VB) and conduction band (CB) of semiconductor 1 are situated at higher energy levels than those of semiconductor 2 (Figure 3b). The alignment of the energy levels facilitates electrons transferring from semiconductor 1 to semiconductor 2, while the holes migrate in the opposite direction. If both semiconductors are in close contact, an effective charge separation will take place upon light illumination. This leads to a decrease in charge recombination and an increase in the lifetime of charge carriers, resulting in higher photocatalyst activity. As one of the typical systems, the VB and CB of MX_2_ are both higher than that of ReS_2_. They can form a Type-II heterojunction structure in which the electrons transfer to ReS_2_ with holes transferred to MX_2_ [21,95,96]. Zhang et al. constructed a type-II heterojunction between ReS_2_ and XS_2_ (X = Mo, W), which can promote the electron transfer to further improve the photocatalytic performance of the counter electrode in dye-sensitized solar cells [96].

Traditional type-II heterojunctions (including n–n junctions and p–n junctions) have several obvious limitations. (1) From the thermodynamic point of view, the improvement of charge separation efficiency is at the expense of oxidation/reduction ability. Specifically, photo-generated electrons will accumulate on the CB of semiconductor 2 with a weaker reduction potential, and photo-generated holes will accumulate on the VB of semiconductor 1 with a lower oxidation potential. Following the type-II charge transfer mechanism, it will inevitably weaken the thermodynamic driving force of the photocatalytic reaction. (2) In terms of dynamics, the transition electrons of semiconductor 2 and the electrons transferred from semiconductor 1 will produce repulsive force. Similarly, the holes between them will also repel each other, which greatly hinders the continuous charge transfer in the type-II heterostructure. (3) Finally, the electrostatic attraction of electrons in semiconductor 2 and holes in semiconductor 1 also render the interfacial charge transfer of type-II composites difficult to achieve [61].

### 3.3. Z-Scheme Heterojunction

Thanks to inspiration from natural photosynthesis, in which two-step photoexcitation and electron transfer are connected in series to form a letter “Z”, similar artificial bionic Z-type heterostructures have been studied and developed in the photocatalysis field [97,98]. The first generation of the liquid-phase Z-scheme system has shuttle oxidation/reduction medium pairs, which are played by electron acceptor/donors, respectively, while there is no physical contact between oxidized semiconductors and reduced semiconductors. However, the first-generation Z-scheme photocatalytic system is limited to use in liquid-phase reactions, severely constraining its practical applications in gas- or solid-phase processes [99]. The second-generation all-solid-state Z-scheme heterojunction uses solid conductors (generally noble metal nanoparticles) instead of redox pairs to make it suitable for both liquid–solid and gas–solid reactions (Figure 3c). After the two semiconductors are excited at the same time, the electrons on the CB of the oxidized semiconductor will migrate to the solid conductor and then transfer to the VB of the reduced semiconductor. The length of this charge transfer path is significantly shortened compared to the initial liquid-phase Z-type heterostructure, which can greatly accelerate the charge transfer and separation efficiency. However, due to the difficulty in accurately controlling the heterojunction structure, the difficulty in directional electron transfer on the interface, and the competition between the light absorption of the mediator itself and the main catalytic components, these two heterostructures have not achieved the ideal photocatalytic effect [100].

The third-generation direct Z-type heterojunction is composed of only a reduced semiconductor and an oxidized semiconductor with a matching band structure and an ohmic contact at the interface (Figure 3d). This ohmic contact interface has certain defects and can be used as a recombination center for the electrons on the CB of the oxidized semiconductor and the holes on the VB of the reduced semiconductor. In addition, the impurity orbits contribute quasi-continuous energy levels and become charge transfer mediators. In addition to inheriting the advantages of the first two generations of Z-scheme systems, the direct Z-scheme without a solid medium greatly reduces the construction cost. Meanwhile, the direct Z-type heterostructure overcomes the light-shielding effect caused by redox pairs or noble metal particles [101]. Although the resistance of the direct Z-scheme at the solid–solid interface is generally considered to be higher than that of the all-solid-state Z-scheme, it is often possible to improve the conductivity of the interface by regulating or optimizing the chemical-binding force through molecular polarization to maximize the advantages of the Z-scheme heterojunction [56].

## 4. Photocatalytic Applications of Heterostructure ReS_2_

The direct band gap of ReS_2_ in both monolayer and bulk form is approximately 1.4 eV [29], allowing for the absorption of light across the entire visible spectrum and into the near-infrared region, making it an ideal material for visible light photocatalysis. The weak interlayer coupling of ReS_2_ results in optoelectronic properties that are largely independent of the number of layers, and the unsaturated edge sites are particularly exposed [35]. These advantages endow ReS_2_ with considerable potential as an efficient photocatalyst or cocatalyst. Considering the limitations of a single photocatalyst, coupling other semiconductors with ReS_2_ into heterojunctions can not only improve the utilization of solar energy but also reduce electron–hole pair recombination, thereby improving redox capacity. ReS_2_-based heterostructure composites have provoked extensive research attention and exhibit excellent photocatalytic activity in the fields of hydrogen production, CO_2_ reduction, and pollutant degradation.

### 4.1. Photocatalytic Hydrogen Production

Hydrogen, which is generated from photocatalytic water splitting, has been regarded as a potential source of energy for the development of a sustainable economy and society. Qualified photocatalysts should have a bandgap over 1.23 eV. The conduction band minimum (CBM) should be higher in energy than the H^+^/H_2_ water reduction potential, while the valence band maximum (VBM) should be lower in energy than the OH^−^/O_2_ water oxidation potential. The band-gap and band-edge positions of monolayer and multilayer ReS_2_ are highly compatible with the energy levels required for water splitting [102]. Pioneer work by Zhang et al. reported that a multilayer structure ReS_2_ with monolayer-like features can generate a tremendous number of trions consisting of two electrons and one hole. This can trigger a two-electron catalytic reaction, offering a PHE rate of 13 mmol g^−1^ h^−1^ under visible light, which is larger than that of most reported TMD composite photocatalysts [33]. Due to its distorted 1T phase structure, transition metal σ bond, and direct band gap, ReS_2_ can serve as an electron reservoir with fast electron transport speed and active electronic states. Consequently, ReS_2_ has been widely recognized as an excellent cocatalyst for constructing a heterostructure with classical photocatalysts such as TiO_2_ [27,34,58,103,104], CdS [44,45,59,62], and g-C_3_N_4_ [46,54,57,92]. It can not only markedly enhance the visible light absorption, charge transfer, and carrier separation efficiency but also offer abundant edge active sites. Additionally, when Re vacancy is produced in ReS_2_, some unsaturated S atoms around the defect sites can promote the adsorption of a H^+^ proton, thus improving the HER performance more than perfect ReS_2_ [34,53,60]. The performance for ReS_2_ as a cocatalyst has been reported to be comparable to widely used noble-metal cocatalysts such as Pt [44,57]. Various photocatalysts have been summarized in Table 1 on the water splitting for the H_2_ production.

The morphology and structure of cocatalysts are important factors that can influence the photocatalytic performance. Cocatalysts typically exhibit morphologies such as nanoparticles, nanospheres, flakes, clusters, nanotubes, and quantum dots. Inspired by nature, Lin et al. designed novel sea-urchin-like ReS_2_ nanosheet/TiO_2_ nanoparticle heterojunctions through a simple one-pot hydrothermal method (Figure 4). Compared with the bulk ReS_2_ cocatalyst composed of a large number of agglomerated nanoparticles or nanosheets, the sea-urchin-like ReS_2_ cocatalyst shows a significantly accelerated charge transfer due to the unusual charge edge-collection effect. The photocatalytic H_2_ evolution rate is 3.71 mmol h^−1^ g^−1^ (an apparent quantum efficiency (AQE) of 16.09%), which is 231.9 times that of P25 TiO_2_ [104].

TMDs with a metallic 1T phase and semi-metallic 1T’ phase generally have high electrical conductivity [106,107]. However, most TMDs with the 1T-phase and 1T’-phase are thermodynamically unstable and can easily convert to the sTable 2H-phase [108,109]. Inspiringly, ReS_2_ has a thermodynamically sTable 1T’-phase crystal structure, which shows great promise for better catalytic performance [32,58]. As cocatalysts, the conductive 1T’-ReS_2_ nanosheets can trigger the host g-C_3_N_4_ nanotubes to generate more photoexcited charges, promote the carrier migration and separation, and supply abundant active sites for photocatalysis. Thus, the 1T’-ReS_2_/g-C_3_N_4_ composite containing 12 wt% 1T’-ReS_2_ exhibits an excellent photocatalytic H_2_ evolution rate of 2275 mmol h^−1^ g^−1^ [57]. Regulating phase structure and constructing an in-plane heterostructure of ReS_2_ can also provide more active sites and increase electron transfer [26]. Chen et al. reported that simply phosphorization treatment led to a shift in the Fermi level due to the extra electrons filling in the d orbital, and thus, ReS_2_ partially transformed from the T’ to the T phase [62]. The developed 2% ReS_2_-CdS/P-0.2 photocatalyst with in-plane T-T’ heterophase junctions enhanced the interfacial effect between CdS and ReS_2_, which can facilitate the electron transfer and exhibit an optimal hydrogen generation rate of 14.68 mmol g^−1^ h^−1^.

Heterostructures with well-identified interfaces are crucial for enhancing photocatalytic performance. The intimate interfacial contact between the individual components via the covalent bond can greatly accelerate the separation of photogenerated electrons and holes. Taking the TiO_2_@ReS_2_ nanocomposites for example, the photoexcited electrons can rapidly transfer to ReS_2_ and diffuse to the active sites due to the chemical interaction of the Ti−O−Re bond [58]. The optimized TiO_2_@ReS_2_ nanocomposites with 30 wt% ReS_2_ exhibit superior photocatalytic H_2_ evolution activity with a rate of 1404 μmol h^−1^ g^−1^. In another 2D/2D heterostructure, the cocatalyst ReS_2_ with abundant active sites is strongly anchored on the TiO_2_ by the chemical interaction of the Ti–S bond, and the established intimate interfacial contact facilitates efficient carrier transfer across the interface [103]. Recently, the strongly coupled interface between atomic-level regulated ReSe_2_ and various semiconductor photocatalysts and the abundant Re/Se active sites have been demonstrated to boost the performance for photocatalytic H_2_ evolution [110].

Chemical modification can adjust the optical and electronic properties of ReS_2_ nanosheets and produce new catalytic properties by increasing the catalytic active sites exposed in solution, which is an important goal for the development of practical and efficient solar hydrogen production devices. Wu et al. used C_6_H_5_COOH on ReS_2_ to form a molecular linkage between 2D ReS_2_ and 0D two-phase TiO_2_ through carboxylate (Figure 5) [56]. This structure provides a Z-scheme molecular electron channel, while the non-functionalized ReS_2_ provides an unstable and inefficient type-II path. The ReS_2_-C_6_H_5_COOH-TiO_2_ composite photocatalyst can not only provide good wettability, connection, and charge transport for solar hydrogen production but also prevent ReS_2_ from being oxidized by photogenerated holes. The composite photocatalyst exhibited high photocatalytic activity for solar hydrogen production (9.5 mmol h^−1^ g^−1^ ReS_2_ nanosheets, 4750 times higher than bulk ReS_2_) and high cycle stability over 20 h.

### 4.2. Photocatalytic CO_2_ Reduction

Anthropogenic emissions of greenhouse gases, such as CO_2_, into the environment can lead to global warming. To reduce CO_2_ production, various strategies have been developed. Using clean renewable energy to fix CO_2_ can promote the carbon cycle and alleviate the global energy crisis. Therefore, photocatalysis has received extensive attention because it can convert carbon dioxide into renewable fuels. Various photocatalysts have been summarized in Table 2 for the CO_2_ reduction.

The valence band edge of ReS_2_ (Ev = 1.36 V vs. NHE) is more positive than the oxidation potential of H_2_O oxidation (0.82 V vs. NHE at pH = 7), while its conduction band edge (E_C_ = −0.20 V vs. NHE) is much lower than the reduction potential of CO_2_ (−0.52 V vs. NHE at pH = 7), indicating that ReS_2_ on its own does not have the ability of photocatalytic reduction of CO_2_ [111]. However, the introduction of defects on the ReS_2_ surface can not only modulate its band structure but also activate the inert surface to realize efficient charge transfer [51]. For example, Zhang et al. synthesized a type I heterojunction which was composed of ReS_2_ nanosheets and CdS nanoparticles via a self-assembly approach (Figure 6a–c). ReS_2_ with S vacancies induced by visible-light illumination is beneficial to the chemical adsorption and activation of CO_2_, as well as the electron transfer between CO_2_ and ReS_2_. The optimized ReS_2_/CdS heterostructure exhibits an enhanced photocatalytic CO_2_-to-CO conversion activity of 7.1 μmol g^−1^ accompanied by a prominent selectivity of 93.4% [51].

Owing to the excellent light absorption, large specific surface area, and abundant active sites of ReS_2_, it can serve as an ideal cocatalyst for photocatalytic CO_2_ reduction [51,64,111]. Huang et al. constructed an indirect Z-scheme heterostructure Au-Pt/Cu_2_O/ReS_2_ via chemical reduction and photodeposition (Figure 6d,e). The photocatalytic efficiency of the prepared heterojunction was significantly improved, and the CO yield was 30.11 μmol g^−1^, which was higher than that of pure Cu_2_O quantum dots (22.87 μmol g^−1^). In addition, the selectivity of CH_4_ and CO can be controlled by manipulating the mass ratio of Au/Pt (Figure 6f,g). Because of the high work function and excellent conductivity of Au and Pt, the electron transfer was accelerated between Cu_2_O and ReS_2_ (Figure 6h) [111].

### 4.3. Degradation of Organic Pollutants

The increase in energy consumption associated with the rapid development of modern society has led to a corresponding rise in environmental pollution, particularly water pollution, which poses a serious threat to human health and global ecological balance. To address this issue, advanced energy production and pollution control technologies should be developed. Among them, photocatalysts have demonstrated their efficacy in degrading organic pollutants. Various photocatalysts have been summarized in Table 3. Specifically, due to the nonbonding d electron of the Re atom, the 1T-ReS_2_ phase exhibits a diamond-chain structural distortion caused by the Re−Re bond interaction [112]. Therefore, the transitions of exciton in ReS_2_ are highly polarization-dependent [113]. More electron–hole pairs and a longer carrier lifetime parallel to the diamond-chain direction give rise to a faster photodegradation rate constant up to 12 times greater than that perpendicular to the diamond-chain direction [114].

The construction of a p-n heterojunction can induce a charge diffusion near the interface in nanocomposites until equilibrium is reached at the Fermi level between the two materials. For example, in the TiO_2_@ReS_2_ nanocomposites, this effect will lift the conduction band position of ReS_2_ higher than that of TiO_2_, leading to a better electron−hole separation efficiency. Under sunlight irradiation, the degradation activity of organic pigments can be significantly improved compared with pure TiO_2_ nanoparticles [47]. Similar strategies take effect such that the ReS_2_/MIL-88B(Fe) heterojunction can extend the lifetime of photoexcited electron−hole pairs, absorb more visible light, and enhance persulfate activation, leading to a better catalytic capacity for ibuprofen degradation compared with the individual component [52]. Electrostatic attraction is significant for fabricating heterojunctions. For instance, the g-C_3_N_4_ and defect-rich ReS_2_ attain close contact, resulting in faster generation of the reactive oxygen species than pristine g-C_3_N_4_ [116].

It is an efficient strategy to improve the photocatalytic performance by constructing a dynamic built-in electric field in a “self-driven” manner to continuously drive the separation of carriers [117]. Recently, a BaTiO_3_@ReS_2_ piezo-assisted photocatalysis heterostructure has been successfully synthesized by Liu et al. [115] through the coupling of interfacial covalent bonds and piezotronics, showing high degradation efficiency for refractory pollutants. The introduction of distorted 1T-ReS_2_ can enhance the light absorption of the hybrid nanostructure. Moreover, the formed interfacial Re-O covalent bond (Figure 7a,b) can serve as the channels for charge carriers’ transfer. In addition, the internal polarization field in the Schottky interface can reduce the Schottky barrier of photogenerated charge transfer. Consequently, the BaTiO_3_@ReS_2_ system obtained high O_2_ activation activity and an excellent synergistic effect under ultrasonic and light conditions. The BaTiO_3_@ReS_2_ exhibits ultrahigh pollutant degradation activity. The apparent rate constant of piezoelectric-assisted photocatalysis is 0.133 min^−1^, which is 16.6 and 2.44 times that of piezoelectric catalysis and photocatalysis, respectively. (Figure 7c,d) [115].

### 4.4. Photocatalytic Reduction of Metal Ions

Heavy metals in aquatic systems and drinking water sources are acutely toxic and carcinogenic to most organisms. Taking Cr (VI) as an example, photocatalytic reduction of Cr (VI) to low-toxic Cr (III) is generally considered to be a promising and practical option because this method is more effective and low-cost and does not produce any hazardous chemicals. Zhou et al. proposed a simple and feasible atomic hybridization strategy to accelerate the reaction kinetics process via constructing electronic pathways [86]. The reduced carbon quantum dots (rCQDs)/ReS_2_ heterostructure is synthesized via the CVD and hydrothermal method (Figure 8a,b). Compared with pure ReS_2_ nanosheets and CQDs/ReS_2_, the reduction reaction rate constants of the pseudo-first-order kinetic model were increased by about 13.1 and 4.3 times, respectively. (Figure 8c). C=O double bonds reduced and then anchored onto ReS_2_ nanosheets can form an electronic channel via Re-5d and O-2p orbital hybridization (Figure 8d), in which a photoinduced carrier of the rCQDs can freely migrate to ReS_2_ for hexavalent chromium reduction (Figure 8e) [86].

### 4.5. Photocatalytic Water Disinfection

The treatment cost of freshwater pollution is high, and the shortage of freshwater resources has become a more complex problem. Photocatalysis techniques have been regarded as one of the most promising strategies for environmental remediation. It is well known that when the photocatalyst absorbs light, electron−hole pairs are formed, which interact with water and dissolved oxygen to generate activated oxygen species (ROS), such as hydroxyl radicals, singlet oxygen, and superoxide [118]. These strong oxidants can destroy essential macromolecules in bacteria, resulting in bacterial disinfection [119]. The field of ReS_2_ photocatalytic sterilization is still in the preliminary stages. Pioneer work by Ghoshal et al. proved that vertically oriented ReS_2_ nanosheets show great potential for photocatalytic water disinfection [78]. However, due to the rapid recombination of the carriers, its inactivation efficiency still needs to be further enhanced.

It has been found that the construction of a 0D/2D heterostructure can greatly improve the sterilization efficiency of a photocatalyst because of the effective separation of charge carriers. Recently, Zhang et al. have reported that a Ag-modified ReS_2_ heterostructure grown on a series of substrates can be used as a visible-light-driven photocatalyst for highly efficient water sterilization. Different substrates and the size of Ag nanoparticles are critical to the photocatalytic bactericidal performance (Figure 9a,b). The Ag nanoparticles on the surface of ReS_2_ nanosheets can not only act as electron capture agents to promote the separation of photogenerated carriers but also inhibit the recombination of photogenerated carriers (Figure 9c). The main active substances in the photocatalysis process are h^+^, •OH, and •O^2−^. All *Escherichia coli* can be inactivated within 30 min at a concentration of 1.72 mg L^−1^ by the reusable Ag/ReS_2_ heterostructure under visible light irradiation (Figure 9d) [85].

## 5. Summary and Outlook

Two-dimensional ReS_2_, a special new material in the family of TMDs, has great potential as a photocatalyst due to its favorable band structure, which is independent of the number of layers, as well as its high specific surface area and numerous active sites. This review summarizes the basic properties and heterostructure synthesis methods of ReS_2_. Several types of ReS_2_-based heterostructures and their recent progress in photocatalysis fields, such as photocatalytic hydrogen production, CO_2_ reduction, and pollutant degradation, are then reviewed, demonstrating that ReS_2_ is an efficient and stable photocatalyst. However, the research on ReS_2_ is still in its early stages, and there is room for improvement of its photocatalytic performance. Based on synthetic chemical mechanisms, semiconductor physics of photocatalysis, and recent research results, the following aspects can be further explored for the rational design of efficient heterostructure photocatalysts:Developing new synthetic methodologies to achieve precise synthesis of ReS_2_ and its composites. Mechanical mixing is usually used in the construction of heterojunctions for ReS_2_, but precise design for the interactions between the two or more materials has not yet been realized in the preparation of photocatalysts, resulting in random morphology and exposed crystal faces of the composites. With the growth mechanism of ReS_2_ as a basis, rational control of the growth conditions is expected to precisely control the size and crystal face orientation. In addition, reducing the size of ReS_2_ down to sub-10-nanometers is another intriguing direction due to the enhanced quantum confinement effect and derived novel photonic properties; however, structural controllability including particle diameter, edge structure, phase transition, etc. is highly important and is also well deserving of more research attention.Surface engineering strategies toward tailorable physicochemical properties of ReS_2_. Few studies on the design and modification of the surface structure of ReS_2_ have been reported for photocatalytic research. Surface defects can become centers of the electron−hole complex, but they are not entirely inutile. For example, introducing Re vacancies in a reasonable way can enhance the adsorption ability of H^+^. In addition, organic molecules can form chemical bonds on the surface S vacancies, which can improve the interface wettability of ReS_2_ on the one hand and fine-tune its energy band position on the other hand.Combining novel characterization methods with theoretical calculations. The surface and interface structure changes in photocatalysis should be thoroughly investigated from both microscopic and transient aspects. Real-time monitoring of intermediates and catalytic products is essential for understanding the photocatalytic mechanism and further optimizing the performance of photocatalysts. For instance, in situ Fourier Transform Infrared spectroscopy and online mass spectrometry can probe the source of hydrogen and the fate of the sacrifice reagents in photocatalytic hydrogen evolution [59]. Ultra-high spatial and temporal resolution technique, such as tip-enhanced Raman spectroscopy, can greatly improve the signal-to-noise ratio and spatial resolution, allowing for the characterization of single molecules and even single chemical bonds. Moreover, the theoretical simulation of the model systems, particularly first-principles, external field simulations, and micro-reaction dynamics simulations, is essential for exploring the fundamental mechanisms of photocatalysis. The binding energy between various intermediates and catalysts can be obtained by calculating the adsorption energy and charge density difference, further reflecting the configuration of the reaction intermediates on the catalyst surface.

Overall, ReS_2_ has the potential to excel as a photocatalytic material, and further investigation into its photocatalytic performance and mechanism could propel the development of photocatalysis and provide new insights into environmental and energy issues.

## Figures and Tables

**Figure 1 molecules-28-02627-f001:**
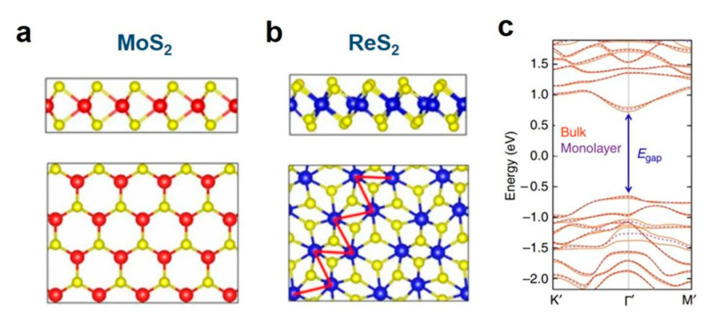
Top view and side view of (**a**) MoS_2_ and (**b**) ReS_2_. (**c**) The electronic band structure of bulk (orange solid curves) and monolayer (purple dashed curves) ReS_2_, as calculated by DFT, is predicted to be a direct bandgap semiconductor with a nearly identical bandgap value at the Γ point [29]. Reproduced with permission: Copyright 2014, Nature Publishing Group.

**Figure 2 molecules-28-02627-f002:**
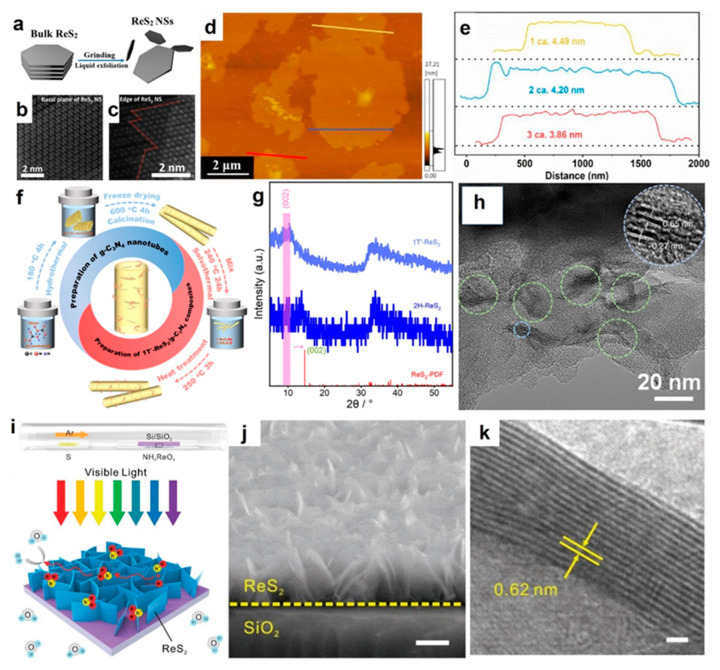
(**a**) Liquid exfoliation of ReS_2_ nanosheets from bulk material [47]. Reproduced with permission: Copyright 2019, IOP Publishing. Atomic-resolution HAADF-STEM images of (**b**) basal plane and (**c**) edge of ReS_2_ nanosheets [50]. (**d**) AFM image and (**e**) thickness profiles of ReS_2_ nanosheets [54]. Reproduced with permission: Copyright 2021, Wiley-VCH Verlag GmbH & Co. KGaA. (**f**) Schematic illustration of the solvothermal synthesis process of g-C_3_N_4_ nanotubes and 1T’-ReS_2_/g-C_3_N_4_ composites [57]. Reproduced with permission: Copyright 2022, ELSEVIER. (**g**) XRD patterns of 1T’-ReS_2_ and 2H-ReS_2_ after heat treatment [57]. (**h**) TEM images of 1T’-ReS_2_/g-C_3_N_4_ composite (12 wt%) [57]. Reproduced with permission: Copyright 2022, ELSEVIER. (**i**) Schematic diagram of the CVD-synthesized ReS_2_ and the two-electron photocatalytic reaction [33]. (**j**) SEM image of the lateral view of the ReS_2_ nanowalls on the silicon substrate; scale bar: 200 nm [33]. (**k**) HRTEM characterization of a ReS_2_ NW with clear lattice fringe; scale bar: 2 nm [33]. Reproduced with permission: Copyright 2018, Wiley-VCH Verlag GmbH & Co. KGaA.

**Figure 3 molecules-28-02627-f003:**
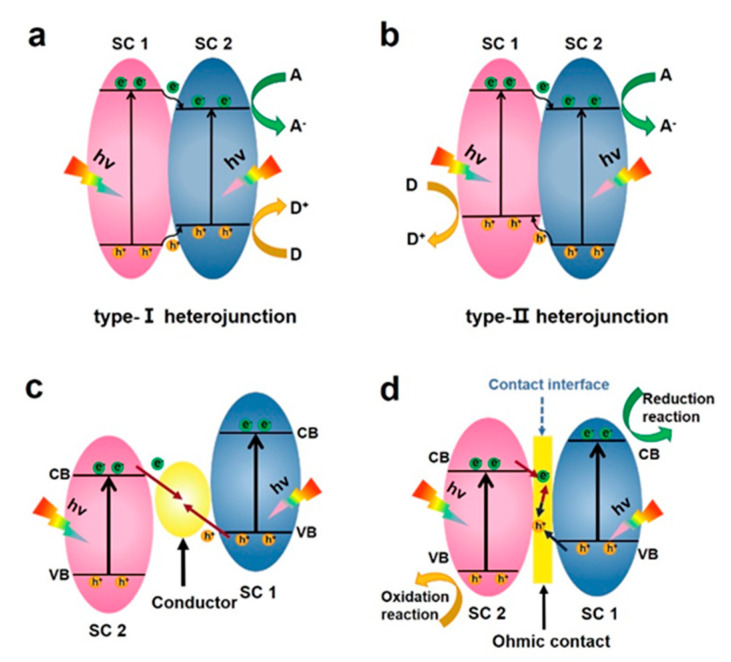
Diagram of (**a**) Type-I, (**b**) Type-II, and (**c**) all-solid-state Z-Scheme heterojunction. (**d**) Direct Z-scheme heterojunction.

**Figure 4 molecules-28-02627-f004:**
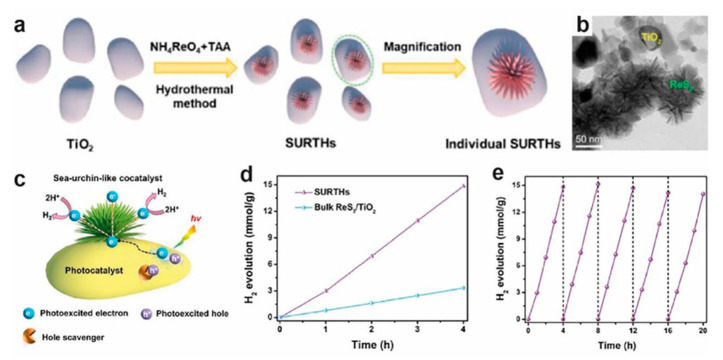
(**a**) The formation procedure of ReS_2_ nanosheet/TiO_2_ nanoparticle heterojunctions (SURTHs). (**b**) TEM images of SURTHs (**c**) Schematic of the sea-urchin-like cocatalyst with the charge edge-collection effect loaded on the photocatalyst for H_2_ evolution. (**d**) Time-dependent photocatalytic hydrogen production of Bulk ReS_2_/TiO_2_ and SURTHs under simulated solar-light irradiation. (**e**) Cycling tests for photocatalytic hydrogen production over SURTHs under simulated solar-light irradiation [104]. Reproduced with permission: Copyright 2022, ELSEVIER.

**Figure 5 molecules-28-02627-f005:**
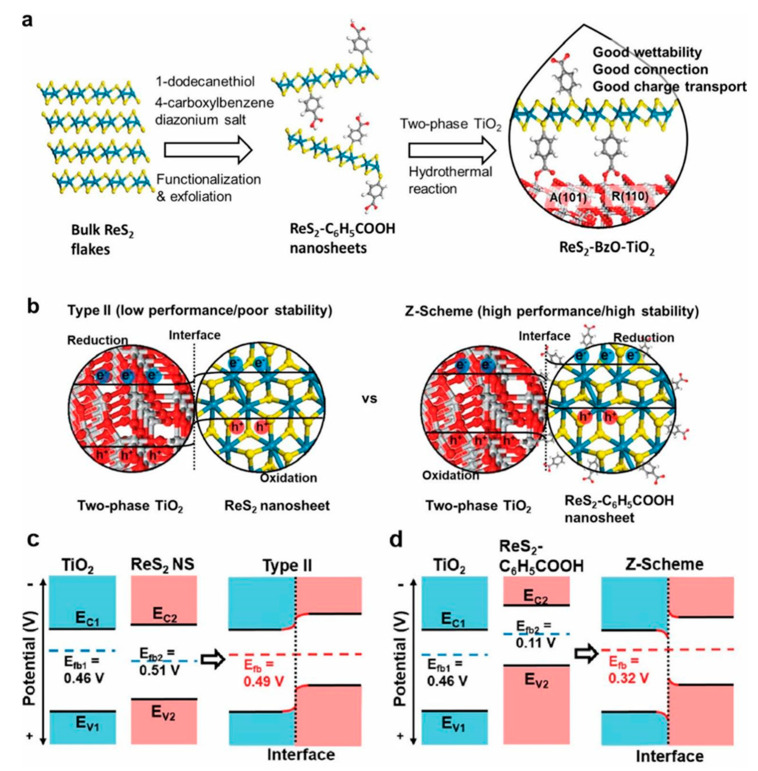
(**a**) Preparation for multiple junctions between functionalized ReS_2_ nanosheets and two-phase TiO_2_. (**b**) Proposed transfer paths of photogenerated charges in terms of contact properties at the interface. Type II: The energy bands of TiO_2_ and nonfunctionalized ReS_2_ nanosheet can bend upward and downward toward the physically contacted interface, respectively. Z-scheme: The energy bands of TiO_2_ and ReS_2_−C_6_H_5_COOH can bend downward and upward toward the chemically contacted interface, respectively. (**c**,**d**) Proposed energy band diagrams for ReS_2_ NS/TiO_2_ (type II) and ReS_2_−BzO−TiO_2_ (Z-scheme) based on the flat-band potentials of each component (*E*_C_—conduction band energy; *E*_V_—valence band energy; *E*_fb_—flat-band energy) [56]. Reproduced with permission: Copyright 2020, American Chemical Society.

**Figure 6 molecules-28-02627-f006:**
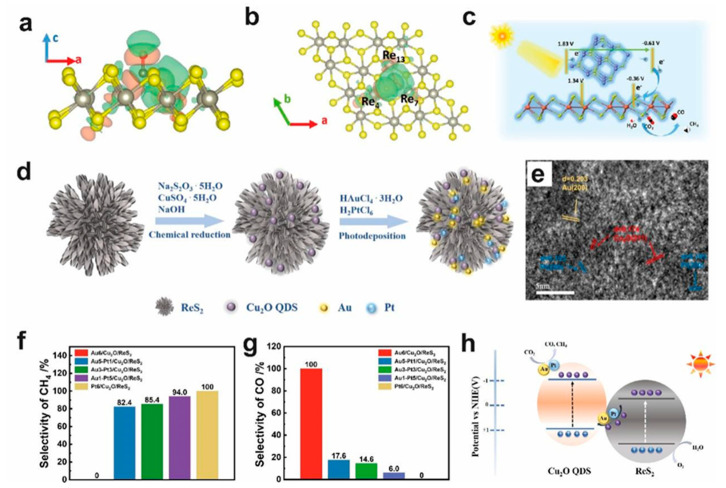
(**a**) Side-view (elevation) and (**b**) top-view (plan) of the electron density distribution of absorbed CO_2_ on Vs-ReS_2_. (**c**) Schematic of photocatalytic CO_2_ reduction in CR12 system under visible-light illumination (λ ≥ 420 nm). The purple-, red-, yellow-, orange-, white-, and black-colored spheres denote Cd, O, S, Re, H, and C atoms, respectively [51]. Reproduced with permission: Copyright 2021, Wiley-VCH Verlag GmbH & Co. KGaA. (**d**) Schematic representation of Au-Pt/Cu_2_O/ReS_2_ synthesis via the approaches of chemical reduction and photodeposition. (**e**) HRTEM images of Au-Pt/Cu_2_O/ReS_2_. Selectivity of CH_4_ (**f**) and CO (**g**) of Au-Pt/Cu_2_O/ReS_2_ with different Au-Pt mass ratios. (**h**) Schematic diagram of the transfer processes of the photoexcited electrons and holes and the mechanism of the photocatalytic CO_2_ reduction [111]. Copyright 2022, ELSEVIER.

**Figure 7 molecules-28-02627-f007:**
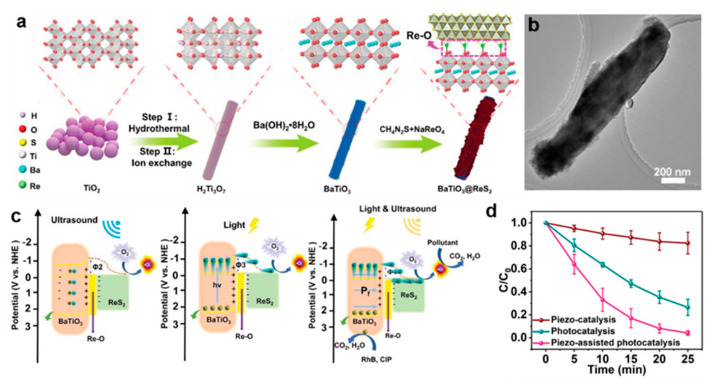
(**a**) Schematic illustration of BaTiO_3_@ReS_2_ system preparation process. (**b**) TEM image of BaTiO_3_@ReS_2_. (**c**) Comparison of the piezo-catalytic, photocatalytic, and piezo-assisted photocatalytic degradation of RhB in the presence of BaTiO_3_@ReS_2_. (**d**) Schematic illustration of the synergy of photocatalysis and piezotronics effects on piezo-assisted photocatalysis system [115]. Reproduced with permission: Copyright 2022, Wiley-VCH Verlag GmbH & Co. KGaA.

**Figure 8 molecules-28-02627-f008:**
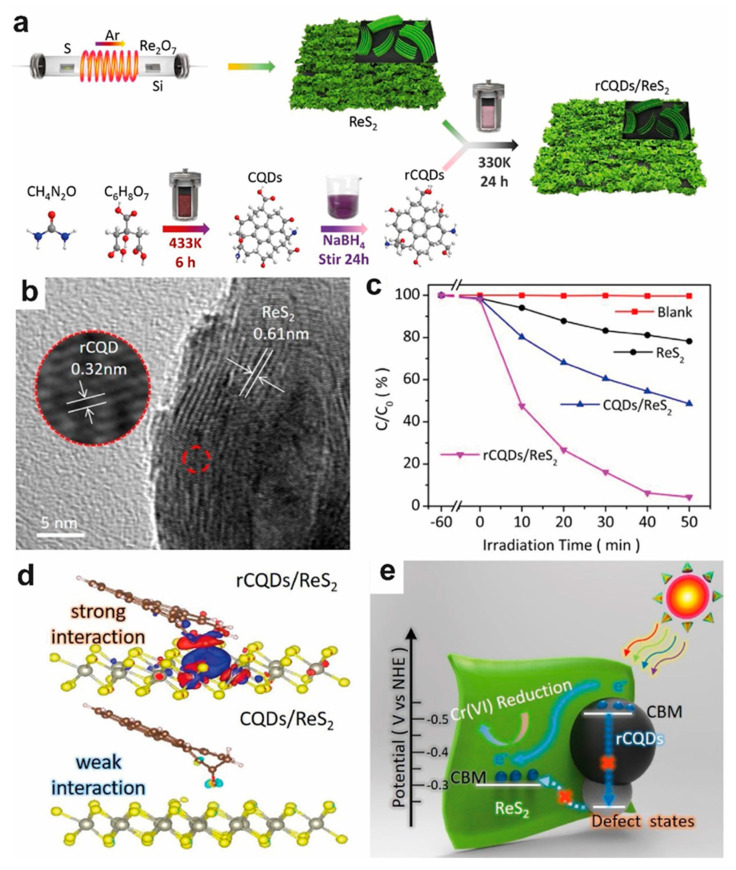
(**a**) Schematic illustration of the preparation process. (**b**) The HR-TEM image of rCQDs/ReS_2_ heterojunctions. Inset: The corresponding TEM image. (**c**) Reduction curves of the Cr (VI) aqueous solutions containing different photocatalysts under simulated sunlight irradiation: no catalyst, ReS_2_ nanosheets, CQDs/ReS_2_ heterojunctions, and rCQDs/ReS_2_ heterojunctions. (**d**) The charge density difference of rCQD(CQD)/ReS_2_ heterojunctions. (**e**) Schematic presentation of the carrier transfer process in the rCQD(CQD)/ReS_2_ heterostructure during reduction [86]. Reproduced with permission: Copyright 2022, ELSEVIER.

**Figure 9 molecules-28-02627-f009:**
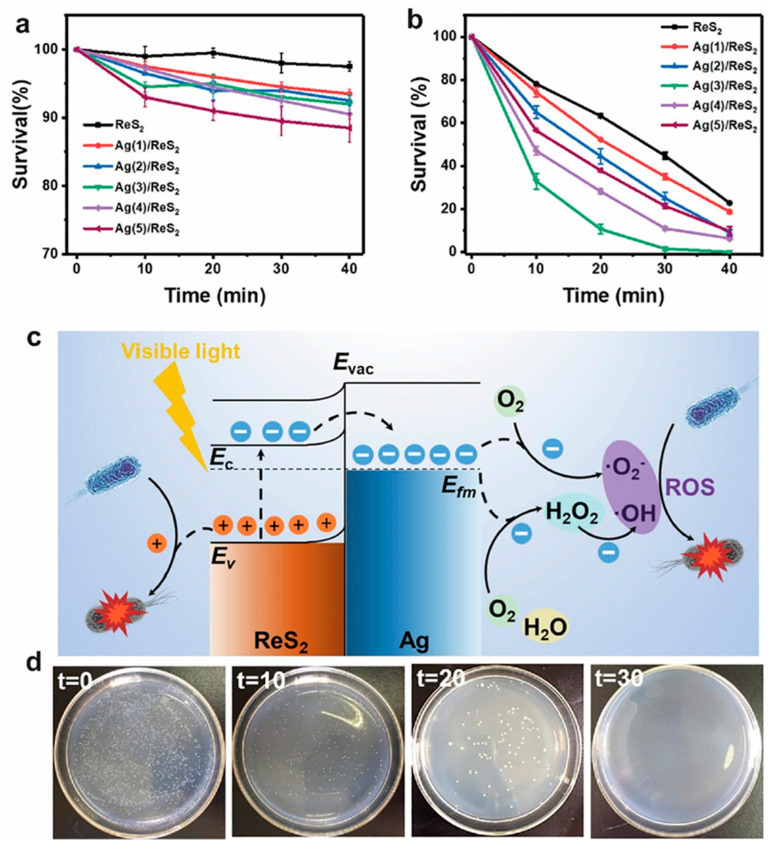
Photocatalytic sterilization efficiency to *Escherichia coli* of different Ag-content samples grown on porous carbon cloth substrate, (**a**) under dark condition, (**b**) under visible light. (**c**) Schematic illustration of charge separation and photocatalytic disinfection via reactive oxygen species (ROS). (**d**) Photos of *Escherichia coli* colonies corresponding to photocatalytic sterilization for 0 min, 10 min, 20 min, and 30 min [85]. Reproduced with permission: Copyright 2022, ELSEVIER.

**Table 1 molecules-28-02627-t001:** Applications of ReS_2_-based heterostructure for hydrogen production.

Photocatalyst	Source of Light	Morphology	Performance	Ref
ReS_2_/ZnIn_2_S_4_	300 W Xenon arc lamp (λ ≥ 420 nm cutoff filter)	cobblestone structure with particles	1858.6 μmol h^−1^ g^−1^	[61]
ReS_2_ nanowalls	300 W Xenon arc lamp (λ ≥ 420 nm cutoff filter)	ReS_2_ nanowalls	13 mmol h^−1^ g^−1^	[33]
TiO_2_/ReS_2_	300 W Xenon arc lamp	ReS_2_ nanosheets and TiO_2_ nanofibers	1404 μmol h^−1^ g^−1^	[58]
ReS_2_/TiO_2_	Solar simulator (λ ≥ 300 nm)	sea-urchin-like structured	3.71 mmol h^−1^ g^−1^	[104]
ReS_2_-BzO-TiO_2_	Solar simulator	nanosheets	9.5 mmol h^−1^ g^−1^	[56]
CdS/ReS_2_	300 W Xennon lamp (λ ≥ 420 nm UV-cutoff filter)	nanorod	137.5 mmol h^−1^ g^−1^	[59]
ReS_2_/Zn_0.5_Cd_0.5_S	visible light irradiation (λ ≥ 420 nm)	nanospheres on ReS_2_ nanosheets	112.10 mmol h^−1^ g^−1^	[105]
ReS_2_/Mn_0.2_Cd_0.8_S	300 W Xenon arc lamp (λ ≥ 420 nm cutoff filter)	cauliflower-like morphology	17.31 mmol h^−1^ g^−1^	[43]
CdS/(Au-ReS_2_)	300 W Xenon arc lamp (λ ≥ 420 nm cutoff filter)	ReS_2_ nanosheets	3060 μmol h^−1^ g^−1^	[45]
g-C_3_N_4_/CdS/ReS_2_	300 W Xenon arc lamp (AM 1.5 G filter)	a hollow spherical nano-shell structure	7141.2 ± 85.7 μmol h^−1^ g^−1^	[44]
ReS_2_/ZnIn_2_S_4_-S_v_	300 W Xenon arc lamp	nanoflower	1.08 mmol h^−1^ g^−1^	[93]
ReS_2_/TiO_2_	300 W Xenon arc lamp	ReS_2_ ultrathin nanosheets	1037 μmol h^−1^ g^−1^	[34]
ReS_2_/g-C_3_N_4_	Solar simulator (AM 1.5G)	nanospheres	1823 mmol h^−1^ g^−1^	[92]
ReS_2_/ZnIn_2_S_4_	Xenon arc lamp (400 nm cutoff light filter)	ReS_2_ nanosheets	2515 µmol h^−1^ g^−1^	[50]
ReS_2_/g-C_3_N_4_	300 W Xenon arc lamp	ultrathin layered 2D/2D structure	3.46 mmol h^−1^ g^−1^	[54]
MoS_2_/ReS_2_@CdS	300 W Xenon arc lamp (λ ≥ 420 nm)	CdS@ReS_2_ nano-spheres and MoS_2_ nanoflakes	171.9 mmol h^−1^ g^−1^	[63]
ReS_2_/TiO_2_	300 W Xenon arc lamp	circle-shaped sheet-like structures 2D TiO_2_	762.3 mmol h^−1^ g^−1^	[103]
ReS_2_/g-C_3_N_4_	visible light irradiation (λ ≥ 420 nm)	ReS_2_ nanoflowers on the surface of g-C_3_N_4_	249 μmol h^−1^ g^−1^	[46]
ReS_2_-CdS/P-0.2	300 W Xenon arc lamp (λ ≥ 420 nm cutoff filter)	CdS nanorods and ReS_2_ nanosheet	14.68 mmol h^−1^ g^−1^	[62]
Ta_3_N_5_/ReS_2_	300 W Xenon arc lamp	CdS nanorods and ReS_2_ nanosheet	615 μmol h^−1^ g^−1^	[53]

**Table 2 molecules-28-02627-t002:** Applications of ReS_2_-based heterostructure for CO_2_ reduction.

Photocatalyst	Source of Light	Morphology	Performance	Ref
ReS_2_@Cu_2_O/Cu	300 W Xennon lamp (λ ≥ 420 nm UV-cutoff filter)	ReS_2_ particles on the surface of Cu_2_O/Cu frameworks	CO (14.3 μmol h^−1^ g^−1^)	[64]
Au-Pt/Cu_2_O/ReS_2_	300 W Xennon lamp	flower-like microsphere	CH_4_ (60.76 μmol h^−1^ g^−1^)	[111]
ReS_2_/CdS	visible-light irradiation (λ ≥ 420 nm)	nanosheets	CO (7.1 μmol h^−1^ g^−1^)	[51]

**Table 3 molecules-28-02627-t003:** Applications of ReS_2_-based photocatalysts of pollutants removal.

Photocatalyst	Type	Synthesis Methods	Morphology	Light Source	Application	Efficiency	Cycle	Ref
ReS_2_/MIL-88B(Fe)	Type-II	solvothermal method	shuttle structure	Both PS and visible light irradiation	Degradation of Ibuprofen (IBP)	100% (3 h)	3	[52]
TiO_2_@ReS_2_	Z-scheme	ultrasonic liquid exfoliation method	ReS_2_ nanosheets	Solar simulator	Degradation of Rhodamine B (RhB)	94% (120 min)	25	[47]
BaTiO_3_@ReS_2_	Type-I	multi-step hydrothermal method	ReS_2_ nanosheets on BaTiO_3_ nanorods	UV-vis light	Degradation of RhB	96% (25 min)	3	[115]
carbon quantum dots (CQDs)/ReS_2_	Type-I	two-step hydrothermal method	rCQDs/ReS_2_ nanosheets	300 W Xenon lamp	Cr (VI)	96% (50 min)	6	[86]
ReS_2_/Graphite Carbon Nitride (CN)	Type-II	electrostatic assembly process	ReS_2_ microspheres and CN nanosheets	300 W Xenon arc lamp (λ ≥ 420 nm cutoff filter)	RhB	94% (30 min)	3	[116]

## Data Availability

Data are contained within the article.

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
