# Peer review of "Photocatalytic Applications of ReS2-Based Heterostructures"

_molecules, 2023, doi:10.3390/molecules28062627_

Round 1
Reviewer 1 Report
A review on the topics of ReS2 photocatalysis is welcome, since the bibliographic research reveals that there is scarce information easily available in the recent literature. The use of rhenium selenide as a material able to capture light is interesting from the perspective of the weak adhesion between the particles, which makes it a peculiar and valuable compound.
The approach about the ReS2 heterostructures is very well explained from the points of view of preparation and activity explanations and as far as I found in literature, it brings a necessary clarity on the subject. Also I appreciate the richness of the information included in the comparative tables gathering the essential from a large number of articles, most of them recently published.
The Conclusions section is consistent and enlightening on the approached topics.
Author Response
Reviewer 1
A review on the topics of ReS2 photocatalysis is welcome, since the bibliographic research reveals that there is scarce information easily available in the recent literature. The use of rhenium selenide as a material able to capture light is interesting from the perspective of the weak adhesion between the particles, which makes it a peculiar and valuable compound.
The approach about the ReS2 heterostructures is very well explained from the points of view of preparation and activity explanations and as far as I found in literature, it brings a necessary clarity on the subject. Also I appreciate the richness of the information included in the comparative tables gathering the essential from a large number of articles, most of them recently published.
The Conclusions section is consistent and enlightening on the approached topics.
Reply: Thanks for the reviewer’s comments.
Reviewer 2 Report
Dear authors,
The article provides an adequate state-of-the-art of the material and a comprehensive overview of recent progress based on ReS2bases heterostructure for photocatalysis. Various growth techniques and the most interesting properties, especially in the perspective of photocatalysis, are effectively reported. The article is well-structured and comprehensive. Conclusions are consistent e references are adeguate. I recommended the publication of thi paper in Molecules after minor revision
row 118-119 : The link between text and figure 2a and b are incorrect.
There is a complete lack of references to figure 2(d-k) in the text
Author Response
Reviewer 2
Dear authors,
The article provides an adequate state-of-the-art of the material and a comprehensive overview of recent progress based on ReS2bases heterostructure for photocatalysis. Various growth techniques and the most interesting properties, especially in the perspective of photocatalysis, are effectively reported. The article is well-structured and comprehensive. Conclusions are consistent e references are adeguate. I recommended the publication of thi paper in Molecules after minor revision
Reply: Thanks for the reviewer’s comments.
row 118-119 : The link between text and figure 2a and b are incorrect.
Reply: The corresponding description has been corrected in the new manuscript with highlights.
There is a complete lack of references to figure 2(d-k) in the text
Reply: The references and the corresponding elaborationin have been supplemented in the new manuscript with highlights.
Reviewer 3 Report
This manuscript summarizes the most recent progress in the ReS2-based heterostructure for photocatalysis. The authors provide a comprehensive overview from the fundamental properties, preparation techniques, combined with a focus on environmental based photocatalysis based on the two-dimensional ReS2 heterostructure. This work shall reveal broad interest towards the research committees encompassing two-dimensional nanomaterials, catalysis, environmental science. Overall, this is a well-organized review and I am happy to recommend the publication of this manuscript in Molecules after minor revisions.
1. An overview chart/figure summarizing the main points of the review shall be even better to provide the readers the basic outlines.
2. In the conclusion and outlook section, the author mentioned that the development of transient, in-situ characterization techniques is crucial for characterizing intermediates in photocatalytic reactions and understanding the reaction mechanism. I agree with this point. Another related work reported by Wong et al. (DOI: org/10.1016/j.apcatb.2019.117897) shall be a good supplement. The in-situ Fourier Transform Infrared spectroscopy and on-line mass spectrometry conducted on the ReS2/CdS heterojunction offer an in-depth understanding for the mechanism of photocatalytic CO2, this work shall be cited and discussed in conclusion and outlook section.
3. In section 4.3, the author reviews the related work of ReS2-based composite catalysts for the degradation of pollutants in detail. Another work of ReS2/g-C3N4 heterojunction for photodegradation (DOI: doi.org/10.1002/cctc.201802021) has been missed. In this work, due to the electrostatic and coordination interactions, the g-C3N4 and defect-rich ReS2 attain close contact to form heterojunctions. It results in faster generation of the reactive oxygen species than pristine CN, as well as increased visible and near-infrared light absorption because of strong photoabsorption of defect-rich ReS2. This work illustrates significance of electrostatic attraction in fabricating heterojunctions and should be supplemented.
4. Recently, Qiao et al. reported atomic-level engineering of defected ReSe2 nanosheets (NSs) can significantly boost photocatalytic H2 evolution on various semiconductor photocatalysts including TiO2, CdS, ZnIn2S4 and C3N4. Although this review focuses on ReS2, based on the similarity of structure and properties, the related work of ReSe2 can also provide theoretical and experimental basis for subsequent heterojunction research. Therefore, this article should be included to Part 4 of this review.
5. The final outlook part needs to be condensed, for example, in-situ characterization and calculation can be combined.
6. There are several typos in the manuscript, like some erroneous use of plural and singular forms of nouns. The author should double check the manuscript.
Author Response
Reviewer 3
This manuscript summarizes the most recent progress in the ReS2-based heterostructure for photocatalysis. The authors provide a comprehensive overview from the fundamental properties, preparation techniques, combined with a focus on environmental based photocatalysis based on the two-dimensional ReS2 heterostructure. This work shall reveal broad interest towards the research committees encompassing two-dimensional nanomaterials, catalysis, environmental science. Overall, this is a well-organized review and I am happy to recommend the publication of this manuscript in Molecules after minor revisions.
1. An overview chart/figure summarizing the main points of the review shall be even better to provide the readers the basic outlines.
Reply: The overview figure has been supplemented as a TOC in the revised manuscript.
2. In the conclusion and outlook section, the author mentioned that the development of transient, in-situ characterization techniques is crucial for characterizing intermediates in photocatalytic reactions and understanding the reaction mechanism. I agree with this point. Another related work reported by Wong et al. (DOI: org/10.1016/j.apcatb.2019.117897) shall be a good supplement. The in-situ Fourier Transform Infrared spectroscopy and on-line mass spectrometry conducted on the ReS2/CdS heterojunction offer an in-depth understanding for the mechanism of photocatalytic CO2, this work shall be cited and discussed in conclusion and outlook section.
Reply: The reference has been added and discussed in our renewed manuscript. Thank you for your constructive suggestion.
3. In section 4.3, the author reviews the related work of ReS2-based composite catalysts for the degradation of pollutants in detail. Another work of ReS2/g-C3N4 heterojunction for photodegradation (DOI: doi.org/10.1002/cctc.201802021) has been missed. In this work, due to the electrostatic and coordination interactions, the g-C3N4 and defect-rich ReS2 attain close contact to form heterojunctions. It results in faster generation of the reactive oxygen species than pristine g-C3N4, as well as increased visible and near-infrared light absorption because of strong photoabsorption of defect-rich ReS2. This work illustrates significance of electrostatic attraction in fabricating heterojunctions and should be supplemented.
Reply: The reference has been added and discussed in our renewed manuscript. Thank you for your constructive suggestion.
4. Recently, Qiao et al. reported atomic-level engineering of defected ReSe2 nanosheets (NSs) can significantly boost photocatalytic H2 evolution on various semiconductor photocatalysts including TiO2, CdS, ZnIn2S4 and C3N4. Although this review focuses on ReS2, based on the similarity of structure and properties, the related work of ReSe2 can also provide theoretical and experimental basis for subsequent heterojunction research. Therefore, this article should be included to Part 4 of this review.
Reply: The reference has been supplemented in the updated manuscript. Thank you for your comment.
5. The final outlook part needs to be condensed, for example, in-situ characterization and calculation can be combined.
Reply: Prospects for in-situ characterization and calculation have been merged and refined. Thank you for your suggestion.
6. There are several typos in the manuscript, like some erroneous use of plural and singular forms of nouns. The author should double check the manuscript.
Reply: Thanks for reviewer’s comments. We have checked the manuscript and corrected the typos and grammar mistakes. The corresponding corrections are marked in cyan in the revised manuscript.